# Inclusion of Soybean Hulls (*Glycine max*) and Pupunha Peach Palm (*Bactris gasipaes*) Nanofibers in the Diet of Growing Rabbits: Effects on Zootechnical Performance and Intestinal Health

**DOI:** 10.3390/ani13020192

**Published:** 2023-01-04

**Authors:** Geovane Rosa de Oliveira, Carla de Andrade, Isabela Cristina Colaço Bez, Antonio Diego Brandão Melo, Vivian Vezzoni Almeida, Washington Luiz Esteves Magalhães, Saulo Henrique Weber, Cristina Santos Sotomaior, Fernando Bittencourt Luciano, Leandro Batista Costa

**Affiliations:** 1Graduate Program in Animal Science, School of Medicine and Life Sciences, Pontifícia Universidade Católica do Paraná (PUCPR), Curitiba 80215-901, Brazil; 2Department of Animal Science, Federal University of Goiás, Goiânia 74690-900, Brazil; 3Embrapa Forest and Nanotechnology, Colombo 83411-000, Brazil

**Keywords:** animal nutrition, nanocellulose, nanotechnology, sustainability

## Abstract

**Simple Summary:**

This study evaluated the inclusion of nanofibers from soybean hulls and pupunha peach palm heart sheaths in the diet of growing rabbits on the performance and health of the gastrointestinal tract. The results reveal that, although the inclusion of 7% Nanopupunha in the diet of rabbits did not alter the performance, it improved intestinal health and increased the lactic-acid bacterial count in the cecum of growing rabbits. Therefore, the inclusion of nanometric particles of plant fibers (<100 nm) in animal feed can be used as a potential fiber alternative and provide excellent physical characteristics for improved nutrient absorption from the diet.

**Abstract:**

This study evaluated the inclusion of nanofibers from soybean hulls and pupunha peach palm heart sheaths in the diet of growing rabbits. Twenty-four New Zealand White rabbits (male and female) were allocated in three experimental groups: control, fed a basal diet; Nanosoy, fed a diet containing 7% soybean-hull nanofibers; and Nanopupunha, fed a diet containing 7% pupunha palm heart-sheath nanofibers. The Nanosoy-group rabbits showed poorer final weight, daily feed intake, and daily weight gain than those in other groups. In the duodenum, villus height, total mucosal thickness, and villus width were higher in rabbits that received nanofiber-supplemented diets than in the controls. Higher villus density and wall thickness were observed in Nanopupunha-fed rabbits than in the controls. In the jejunum, although the crypt depth was higher in Nanosoy-fed rabbits, the villus height:crypt depth ratio was higher in the Nanopupunha-fed group. Nanosoy-fed animals exhibited increased count Enterobacteriaceae populations. Rabbits in both nanofiber-fed groups exhibited higher lactic-acid bacterial counts than those in the control-diet group. Therefore, although the inclusion of 7% Nanopupunha in the diet of rabbits did not alter the performance, it improved intestinal health and increased the lactic-acid bacterial count in the cecum of growing rabbits.

## 1. Introduction

In recent years, several co-products from manufacturing agricultural have been used as ingredients to supplements to traditional animal feed and have prompted the development of new technologies for better utilization by animals, in addition to reducing their negative impact on the environment by discarding co-products. Nanotechnology studies the phenomena and manipulation of materials at the nanoscale and contributes to the use of co-products in the agricultural sector. This has benefitted the sustainable development of animal production and shown promising results in several research areas as food, biomedical, cosmetics, and pharmaceutical [1,2]. Nanofibers are stable atomic structures with biodegradable characteristics and low-production costs [3,4]. They are obtained from renewable natural sources [5] and can improve animal production and food safety.

Nanofibers have long-chain and amorphous cellulose structures which increase their surface area and reactivity [6]. Nanometric-scale fibers have greater potential for absorption, reaction [7,8] and can serve as substrates for the fermentation of volatile fatty acids (VFA) by intestinal bacteria to increase the energy supply in animals. Animals fed a nanofiber (scale: <100 nm) diet show better zootechnical performance and greater productive efficiency [9,10] than those co-products rich in fiber resistant to digestive processes in monogastric animals.

El-Ratel et al. observed higher performance in rabbits that were fed a turmeric-nanoparticle-supplemented diet [11]. Xu et al. observed a 32% improvement in weight gain in piglets fed on a nanochitosan-supplemented diet [12]. The use of a nanomineral (nano-selenium) enriched diet increased live weight and improved feed conversion in broilers [13,14], and increased nutrient digestibility and milk quality in sheep [15,16].

Nanofiber-supplemented diets in mice can also improve intestinal health and intestinal morphology by preventing villus shortening and intestinal injuries [9,10,17]. Another highlighted benefit of this diet is an improved blood–lipid profile [7]. In addition, the inclusion of 0.75 or 1.5% cellulose nanofibers in the murine diet results in no cytotoxic effects on small intestine (SI) epithelial cells [7], which makes them a safe source of animal nutrition.

To the best of our knowledge, no study has been conducted to evaluate the effects of adding nanofibers from soybean hulls (*Glycine max*) and palm heart sheaths (*Bactris gasipaes*) into the diet of rabbits. We hypothesized that the inclusion of nanofibers from soybean hulls and pupunha peach palm sheaths in rabbit diets would have positive effects on the health of the gastrointestinal tract, absorption, and metabolism of nutrients and the intestinal microbiota. Therefore, the present study was conducted to investigate the effects of 7% soybean-hull and 7% palm-heart-sheath nanofiber diets on the zootechnical performance, frequency of diarrhea, relative organ weight, pH of the digestive tract content, structural and ultrastructural histology of the intestine, biochemical, and immunological parameters, and cecal microbiota in growing rabbits.

## 2. Materials and Methods

The present study was conducted at the Rabbit Farming Sector of the Pontificia Universidade Católica of Paraná (PUCPR) for 42 d. The experiment was approved by the Ethics Committee for the Use of Animals, PUCPR, under number 903—2nd version.

### 2.1. Experimental Animals

Twenty-four New Zealand White rabbits (male and female) were weaned at the age of 35 d at an initial body weight (IBW) of 0.932 ± 0.171 kg. The animals were distributed in a randomized block design according to weight, placed in individual cages (80 × 60 × 45 cm), and divided into three treatment groups, with eight animals per group (n = 8). Animals received water and food *ad libitum* throughout the experimental period.

### 2.2. Production of Nanofibers

Nanofibers from soybean hulls and the sheath of the pupunha peach palm were produced through mechanical methods at the Wood Technology Laboratory of the Brazilian Agricultural Research Corporation (EMBRAPA Florestas) located in Colombo/PR. The peach palm and soybean hull samples were fragmented in a 450 W blender for 5 min and subjected to microfibrillation in a microprocessor (Super Masscolloider; Masuko Sangyo Co. Ltd., Kawaguchi-city, Saitama-pref, Japan), which consisted of a rotating disk coupled with a fixed disk containing an adjustable opening where the sample was deposited. The equipment parameters to obtain a gel with a 7% nanofiber concentration were as follows: rotation = 1500 RPM; number of passes = 30; and distance between the discs = 0.1 mm.

The nanofiber was characterized in terms of their dimensions and chemical characteristics. For the analysis, the sample was submitted to a procedure of individualization of the cellulose fibers. For this process, the sample was diluted in 4 parts of PA ethyl alcohol in eppendordf. This diluted sample was subjected to sonication for 60 min. After this step, the sample was dripped onto a screen covered with Parlodion. Once dried at room temperature, the samples were analyzed using a Transmission Electron Microscope (TEM), brand JEOL, model JEM1200EX-II, located at the Center for Electronic Microscopy (CME) of UFPR. The resulting images were processed with the software Paint.net TM version 3.5.10, which allowed for an estimation of the fibril dimensioning (Figure 1).

The bromatological analysis of the vegetable raw materials (soybean hulls, pupunha peach palm sheath) and the respective nanofiber gels were performed at the Nutrition Laboratory at PUCPR and expressed in % based on dry matter (m/m). For weighing the samples, an analytical precision balance was used (Mars, AY220). The total fiber content was determined using the enzymatic-gravimetric method modified according to the Chemical and Physical Methods methodology for food analysis of the Analytical Norms of the Instituto Adolfo Luz [18].

After defibrillation, the gel was stored in a refrigerator with 93% humidity and added to the feed. The inclusion of Nanosoja and Nanopupunha gel with 7% of Nanofibers in the natural matter was calculated to obtain an increment of 7% of nanofibers in the dry matter of the feed, considering the formula: Suspended mass of nanofibers (MSN) = Final mass of feed (MF) × percentage of nanofibers in the feed (NF) ÷ percentage of nanofibers in the gel (NG). The gel was composed of nanofibers in suspension, containing the following fiber levels: (1) nanosoy gel (%) contained neutral detergent fibers (NDF) = 5.82 and acid detergent fibers (ADF) = 5.34; (2) nanopupunha gel (%) contained NDF = 9.40 and ADF = 6.76. The feed was then pelleted in an electric machine (Grinder CAF-22 Machines, Rio Claro, SP, Brazil) and dried at 55 °C for approximately 16 h in a forced ventilation oven (Solab, Primar—SL-102/1540) located in the Nutrition Laboratory of PUCPR. Drying was conducted to reduce humidity and to allow greater durability of the product in the animal feed.

### 2.3. Diets

Diets were formulated based on recommendations for the nutritional requirements in growing rabbits [19] (Table 1). Three diets were tested: Control, basal diet; Nanosoy, diet containing 7% nanofibers from soybean hulls; and Nanopupunha, a diet containing 7% nanofibers from the palm heart sheath.

### 2.4. Performance Variables

The performance variables, final body weight (FBW), daily feed intake (DFI), daily weight gain (DBWG), and feed conversion (FC) were calculated from the individual weight of the animals weighed at the beginning and end of the experiment and from the quantification of the feed consumed and left in the trough. The occurrence and frequency of diarrhea was recorded daily according to the methodology adapted from a previous study [20].

### 2.5. Relative Organ Weight

At the end of the experiment, all animals were euthanized to measure the relative weights of their digestive tract organs (stomach, liver, SI, and large intestine (LI)), kidneys, and spleen. The relative weights of the organs were calculated according to the final body weight of the rabbits.

### 2.6. Determination of the pH of the Stomach, Jejunal, and Cecal Contents

Immediately after the animals were euthanized, the digestive organs (stomach, jejunum, and cecum) were removed to measure the pH of the digesta using a pH meter (HI 99163; Hanna Instruments, RI, USA) following the methodology adapted from Silveira et al. [21]. In the stomach, an incision was made approximately 2 cm from the antropyloric region. In the jejunum, an incision was made in the median portion and the cecum incision was made in the median portion for the pH measured.

### 2.7. Structural Analysis of the Intestinal Epithelium

Immediately after pH measurement, 3-cm-long sections from the duodenum and jejunum were collected and fixed in 10% formaldehyde for 72 h. Subsequently, at the PUCPR Histopathology Laboratory, these samples were exposed to gradual concentrations of alcohol and embedded in paraffin. Using a microtome, 4-µm-thick sections were prepared and stained using the hematoxylin-eosin stain. Slides were analyzed using light microscopy. For structural analysis, the slides were scanned using Axio Scan Z1 equipment (Carl Zeiss AG, Jena, Germany). The measurement of villus height (VH), crypt depth (CD), VH/CD ratio, villus density (VD), villus width (VW), total mucosal thickness (TMT), and wall thickness (WT) were performed using the ZEN software (ZEISS).

### 2.8. Ultrastructural Analysis of the Intestinal Epithelium

Eight samples measuring 0.50 × 0.50 cm^2^ were obtained from the duodenum and jejunum of the animals for ultrastructural analysis. At the time of collection, the samples were carefully washed with saline solution (0.9%), immersed for 1 h in Karnovsky’s fixative solution that contains glutaraldehyde 2.5% (Exodo Casv—111-30-8) and sodium cacodylated 3.424% (013850; Sigma-Aldrich, Saint Louis, MO, USA), cut, and permanently stored in the same solution. For ultrastructural analysis, the samples were fixed for approximately 2 h in 2% osmium tetroxide, dehydrated in acetone, and dried in CO_2_ until they reached the critical point. The samples were sputtered with gold (an electron semiconductor) and examined under a scanning electron microscope (JSM-6010PLUS/LA, Jeol, Tokyo, Japan) at Fundação Oswaldo Cruz (Fiocruz, Curitiba, PR, Brazil) to evaluate the villi structure and villi count per determined area (0.922 mm^2^), with six replicates per treatment.

### 2.9. Blood Biochemical Parameters

On day 1 and 42 of the experiment, 5 mL of blood was collected through cardiac venipuncture for analysis of blood glucose, total cholesterol, and triglycerides. The Accu-Chek Guide kit (Basel, Switzerland) was used for blood glucose analysis. For the evaluation of plasma levels of total cholesterol (Cholesterol SL reagent; ELITechgroup Inc., Sées, France) and triglycerides (Triglycerides Mono; ELITechgroup Inc.), the blood was centrifuged at 5000× *g* for 5 min, and the plasma was collected and transported to the PUCPR Veterinary Hospital and stored at −20 °C until needed for measurement using the automated reader (EL80; ELITech-Clinical Systems, Cairo, Egypt).

### 2.10. Immunology

For the quantification of specific immunoglobulins, rabbit IgG (Immunoglobulin G—ERB0171; Wuhan Fine Biotech Co., Ltd., Wuhan, Hubei, China) and rabbit IgM (Immunoglobulin M—ERB0172; Wuhan Fine Biotech Co., Ltd., Wuhan, Hubei, China) ELISA kits were used. The blood samples were centrifuged at 10,000 RPM for 5 min. The serum was collected and stored at ×20 °C until the tests were carried out at the Laboratório Imunova Análises Biológicas LTDA, PUCPR.

Serum samples were prepared according to the methodology described for rabbit IgG and IgM ELISA kits, which were based on the sandwich technique (linked to an immuno-absorbent enzyme that reacts with the added substrate). The 96-well plates were pre-coated with biotin-conjugated anti-IgG and anti-IgM antibodies that were used as detection antibodies. To perform the tests, the samples for IgG were diluted to 1:10,000 and those for IgM to 1:30,000, according to the curve fit of the kit, which ranged from 0.313–20 ng/mL. The absorbance was measured at 450 nm using an ELISA reader (Epoch2T; BioTek Instruments, Winooski, VT, USA) at the Laboratory of Agro-Food Research and Innovation at PUCPR.

### 2.11. Microbiota in the Cecal Content

Samples of the cecal contents were collected and stored in sterile, hermetically sealed falcon tubes, and were refrigerated on ice for approximately 90 min. For the analysis of the concentrations of the different bacterial groups, approximately 1 g of digesta was weighed and decimal dilutions of the samples were performed in 1% peptone water solution (Kasvi, Conda, S.A., Madrid, Spain).

Colony counts were enumerated using the petri dish pour plate technique on MacConkey agar (Kasvi, Italy) for total Enterobacteriaceae (CBT) and on Man, Rogosa, and Sharpe agar (MRS; Sigma-Aldrich, Saint Louis, MO, USA) for lactic-acid bacteria. The plates were incubated under aerobic conditions for 24 and 48 h at 37 °C. The counts were determined according to the FDA-mandated method [22]. Bacterial concentrations were subjected to log_10_ transformation prior to analysis. After the colonies were isolated, the Enterobacteriaceae kit was used, which consisted of 10 biochemical tests associated with lactose fermentation (from the isolation medium) and allowed for the differentiation of gram-negative bacilli from other oxidase-negative bacteria.

### 2.12. Data Analysis

The data were submitted for testing the adequacy of the linear model using the statistical package Statgraphics 4.1. The initial live weight of the animals was used as a covariate for performance parameters. The averages of performance parameters, digestive content pH, relative organ weight, structural and ultrastructural histology of the intestine, biochemical and immunological parameters, and cecal microbiota of the treatment groups were compared using ANOVA type III, followed by Tukey’s test, and a Levene test whenever homoscedasticity was observed. A Shapiro–Wilk test was conducted to test for data normality. For all analyses, *p* < 0.05 was considered statistically significant. All analyses were performed using SPSS 25 [23].

## 3. Results

### 3.1. Performance

The Nanosoy-diet-fed rabbits were observed to have lower final FBW, DFI, and DBWG (*p* < 0.05) as compared with the Nanopupunha-diet- and control-diet-fed rabbits but showed no difference in FCR (Table 2). The zootechnical performances of the animals in the control and Nanopupunha groups were statistically equal. Diarrhea was not observed in any of the animals throughout the experimental period.

### 3.2. Relative Organ Weight

The relative weight of the stomach of the animals in the Nanopupunha group was observed to be lower (*p* < 0.05) than that in the animals of the control group, and the relative weight of the SI of the Nanopupunha-group animals was higher (*p* < 0.05) than that of the control- and Nanosoy-group animals. The relative weight of the LI was lower in the Nanopupunha-diet-fed animals than that in the animals of the other groups. Liver and spleen relative weights were lower (*p* < 0.05) in Nanosoy-fed animals, with no weight difference between the control and Nanopupunha groups (Table 3).

### 3.3. Determination of the pH of the Stomach, Jejunal, and Cecal Contents

The Nanosoy-fed animals showed a reduction (*p* < 0.05) in the pH of the stomach and jejunal contents. The pH of the cecal contents was higher (*p* < 0.05) in the Nanopupunha-fed animals when compared with that in the Nanosoy-fed animals and showed no difference with that in control-diet-fed animals (Table 4).

### 3.4. Structural and Ultrastructural Analyses of the Intestinal Epithelium

For the duodenum, higher (*p* < 0.05) villus height (VH), villus width (VW), and total mucosal thickness (TMT) were observed in the nanofiber-fed rabbits compared with those in the control-diet-fed animals. Greater crypt depth (CD) (*p* < 0.05) was observed in Nanosoy-fed rabbits compared to that in animals in the control group, and greater (*p* < 0.05) villus density (VD) and wall thickness (WT) were observed in the Nanopupunha-diet-fed animals compared with that in the control-diet-fed group. For the jejunum, the CD was higher (*p* < 0.05) in animals fed the Nanosoy diet than in the animals in the control group. A higher (*p* < 0.05) VH:CD ratio was observed in animals fed the Nanopupunha diet than in those fed the Nanosoy diet (Table 5).

Ultrastructural analysis of the duodenum of rabbits (Figure 2) revealed that the villi appeared flatter and worn plate-like in the Nanosoy-fed animals, and in the form of a tongue, with folds, for the animals in the control group. In all groups, overlapping and folding of the villi in the duodenum was observed. The villi of the jejunum (Figure 3) were thinner and tongue-shaped, with a greater density. However, a greater apical loss of the epithelial lining cells was observed in the villi of Nanosoy-fed animals.

### 3.5. Biochemical Parameters

Blood glucose and triglyceride levels in the animals at the end of the experiment were not influenced (*p* > 0.05) by the inclusion of nanofibers in the diets. Only the control diet showed a significant increase (*p* < 0.05) in blood glucose levels from day 0 to 42 of the experiment. However, a lower total cholesterol rate (*p* < 0.05) was observed in Nanosoy-fed rabbits than in the Nanopupunha-fed group (Table 6).

### 3.6. Immunology

In the present study, there was no difference (*p* > 0.05) in the concentrations of IgM and IgG in the rabbits that received different treatments (Table 7).

### 3.7. Microbiota in the Cecal Content

The Enterobacteriaceae count in the cecal content of rabbits was higher (*p* < 0.05) in the Nanosoy-diet-fed animals than that in the other groups. However, the lactic-acid bacterial count was higher (*p* < 0.05) in the groups that received diets containing nanofibers (Figure 4).

## 4. Discussion

### 4.1. Zootechnical Performance

Rabbits fed a Nanopupunha-supplemented diet had better zootechnical indices (*p* < 0.05) than those who were fed a Nanosoy-supplemented diet, and had similar indices to the control-group animals. This indicated that the inclusion of 7% Nanopupunha in the diet had good diet acceptance in the animals. Good acceptability, although without changes in performance, was also observed by Andrade et al. [24], who studied the inclusion of up to 21% nanofibers from palm heart sheath into the diet of rats. The Nanopupunha-included diets promoted higher FW in the animals (2.770 kg), which was an increase of 17.81% in weight in comparison to the FW of 2.291 kg recorded with the Nanosoy diet.

Other studies have also shown that nanoparticles improve animal performance and productivity. Xu et al. [12] supplemented piglet diets with chitosan nanoparticles (400 mg) and found an increase of 38.82 g in DWG. The use of nanominerals such as nano-selenium increases the BW of broilers [13,16] and sheep [15], whereas those such as nano-zinc improve the quality of milk, eggs, and meat [25,26]. In this study, the nanofibers of pupunha peach palm were found to be an excellent alternative for partial replacement of fibers in the diet of growing rabbits.

The peach palm nanofibers have lower amounts of lignin and pectin and higher amounts of cellulose and hemicellulose [27]. This composition improves the process of obtaining nanofibers from the plant and indicates the important role these nanofibers can play in the rate of passage of food when included in the diet. Diets with a higher content of insoluble non-starch polysaccharides (NSPs) act as substrate sources for commensal microbiota which improve the synthesis and absorption of short-chain fatty acids (SCFA), which [28] in growing rabbits. The presence of glucose and xylose bound to the cellulose matrix improves the quality of the peach palm nanofibers [27]. Nanoparticles can improve animal performance as they have greater availability, reactivity, and potential to improve the digestion and absorption of nutrients in the gastrointestinal tract because of their greater ability, than those fibers that interact with that environment.

Dietary fiber benefits digestibility, gut health, and commensal gut microbiota in monogastric animals [29]. The worst performance results observed with the inclusion of Nanosoy diets may have occurred because of the composition of the cell wall NSPs of soybean hulls. Studies have shown that soluble NSPs present in soybean hulls increase digesta viscosity, reduce digestibility and nutrient absorption, and limit growth in monogastric animals [30,31]. Other antinutritional factors found in soybean hulls are oligosaccharides (stachyose and raffinose) and protease inhibitors, which reduce the digestibility of amino acids, minerals, and lipids [32,33,34], thereby causing intestinal disorders and a decrease in animal performance.

### 4.2. Relative Organ Weight

In comparison to that in the Nanosoy-fed group, the inclusion of Nanopupunha in the diet resulted in a higher FW in the animals because of the greater development of the SI in growing rabbits and an improvement in the digestive capacity of the diet. Changes in the SI of rabbits can be caused by the inclusion of fiber in the feed, which increases the weight and digestive capacity of the gastrointestinal tract [35]. The inclusion of insoluble fiber in the diet, such as Nanopupunha, helps in the development of the SI and absorption of nutrients through greater cell proliferation induced by an increase in the synthesis of SCFA; mainly butyric acid [36]. These changes depend on the chemical structure of the fiber, its cellulose content, the degree of its lignification, and the physiological state of the animal [37].

In our study, the inclusion of 7% Nanosoy in the diet resulted in a greater weight of the LI compared to that in the Nanopupunha-fed animals. Arruda et al. [38] reported higher cecal weight and content in animals fed soybean hulls. The soybean hull fibers—pectin and oligosaccharides—reduce digestion of the diet in the SI, thereby contributing to an increase in the volume of digesta in the cecum [31,34,38]. This results in the greater development of the cecum in rabbits because of an increase in excreta production and a higher rate of microbial fermentation.

Animals fed Nanosoy had lower relative weights of the spleen and liver than those in the animals of the other groups. According to Ewuola [39], changes in the spleen can be induced by cellular destruction caused by toxic substances and/or antinutritional factors; this results in reduced splenic weight. The presence of indigestible oligosaccharides in the SI of monogastric animals modifies the intestinal microbiota and SCFA synthesis, resulting in the production of CO_2_ and H_2_ [40], which induces cell death and decreases the proliferation of B cells in the intestine and spleen [29,41]. This can interfere with splenic function (transformation and transportation of intestinal fluids), causing digestive and nutritional disorders [42] and resulting in a decrease in splenic weight [30].

The lower hepatic weight in the Nanosoy-fed animals was due to the chemical structure of the soybean hull fibers, which promoted a lower absorption capacity of the dietary and negatively influenced the liver weight in growing rabbits. Soluble fibers in the soybean hull stimulate an increase in digesta viscosity, which reduces the action of lipase [43], leading to changes in lipid absorption and metabolism [32,34]. These factors affect triglyceride content in the liver [44] as well as cholesterol levels (Table 6) in the blood serum, which may favor lipogenesis in adipocytes.

In addition to the synthesis of plasma proteins and clotting factors, the liver is also responsible for essential metabolic and biochemical functions [45]. However, basic liver functions can be altered by dietary factors, such as the composition of fibers present in the diet, and hormonal factors [46,47], which reduce liver weight and worsen performance outcomes.

### 4.3. Determination of the pH of the Stomach, Jejunal, and Cecal Contents

The stomach (2.14) and jejunal (6.88) contents of the animals fed with Nanosoy had a lower pH compared with that in animals fed on other diets, and the cecal contents of the animals fed with Nanosoy had a lower pH (6.47) compared to that in animals fed with Nanopupunha. Arruda et al. [38] reported lower pH (6.10) in the cecal content of rabbits fed a diet containing soybean hulls. Soybean hulls have a high content of soluble NSPs, which increases the viscosity of the intestinal contents and reduces the absorption of glucose in the SI. This increases the fermentation rate of acetate and butyrate in the cecum [48] and results in higher cecal acidity. Soy-soluble polysaccharides can affect the digestibility of other nutrients in the diet [31] and cause changes in the physiology of the gut ecosystem.

The pH value of the cecal content in Nanopupunha-diet-fed rabbits is a good indication that a diet with Nanopupunha improves cecal fermentation and intestinal health. Thus, maintaining a higher pH in the LI [28] protects the intestinal epithelium against pathogens as the protective mucus layer of the digestive tract and LI mucins are less acidic and have low resistance to proteolysis, which aids microbial activity [49]. However, in addition to the fiber’s source, its composition and particle size can also contribute to an improvement in the SCFA profile, prevent the risk of intestinal disorders [50], and maintain a pH close to neutrality. These factors can promote favorable conditions for microbial fermentation, increase digestive efficiency, and increase the use of nutrients in the diet of growing rabbits.

### 4.4. Structural and Ultrastructural Analyses of the Intestinal Epithelium

The intestinal epithelial wall functions as a protective and selective barrier for the entry of nutrients and substances that can be harmful to animals [51]. Therefore, fiber diets that promote greater absorption of nutrients can improve the health of the SI mucosa [52] and intestinal function [53]. This makes the use of nanofibers attractive, owing to their ability to improve barrier function and adhere to the intestinal epithelium, while reducing possible damage to the villi.

Regarding the structural analysis of the intestinal epithelium, a higher AV was observed in the duodenum of animals fed with nanofibers (Nanosoy = 939.60 μm and Nanopupunha = 964.77 µm). Chao and Li [54] found a higher AV (734 μm) in the duodenum using a diet with 15% peanut husk and 7% wheat straw.

The Nanosoy diet increased the CP in the duodenum (172.30 μm) and jejunum (139.75 μm) of the rabbits. This increase in CP may be indicative of the high rate of proliferation of crypt cells due to the high rate of cell renewal of the intestinal epithelium. This causes a deterioration in performance because of the greater expenditure of energy for cell renewal and absorption of nutrients. In addition to increasing intestinal viscosity, soluble NSPs have antinutritional effects and low efficiency in the use of nutrients. This may result in changes in the dynamics of the intestinal physiology, indicating a reduction in DNA concentration and cell death in the SI, which leads to an increase in CP [32].

Rabbits are adapted to high-fiber diets. However, the efficiency of fiber diets depends on the best intestinal health condition, as well as the supply of fibers that help to improve the absorption of nutrients and lower energy losses. Therefore, the lower CD, higher VD, and higher VH:CD ratio obtained with Nanopupunha supplementation suggest the greater ability of nanofibers to interact with the intestinal mucosa and improve nutrient assimilation by increasing the surface area of the intestinal cells.

In the ultrastructural analysis of the intestinal epithelium, the Nanopupunha-diet-fed rabbits showed more intact villi in the duodenum and jejunum compared to those in the Nanosoy-diet-fed animals in whom the villi were observed to have suffered a great loss of cellular material. The most abrasive effects on the intestinal mucosa of rabbits occurred because of the higher lignocellulose content of the soybean hulls [55]. Some thermostable antinutritional factors are present in soybeans, such as protease and oligosaccharide inhibitors [56], especially raffinose and stachyose [33]. These factors, in combination with increased intestinal viscosity, an increased rate of cell turnover in the mucosa, noticeable cell loss, and loss of surface area [32,57], sufficiently impaired intestinal absorption in Nanosoy-fed rabbits.

### 4.5. Blood Biochemical and Immunological Parameters

The blood glucose and triglyceride levels of the animals fed a nanofiber-supplemented diet were not altered significantly, which indicated the adequate maintenance of homeostasis during the experimental period. According to Quesenberry and Carpenter [58] and Kahn [59], blood glucose levels in fasting rabbits range from 75–155 mg/dL. These levels were lower than those detected in control- (169.50 mg/dL) and Nanosoy-diet-fed animals (191.62 mg/dL) and similar to those in Nanopupunha-diet-fed animals (153.12 mg/dL) in our study. In a study by Andrade et al. [24], the inclusion of peach palm nanocellulose in the diet did not significantly alter blood glucose levels in rats, which indicated that nanofibers had a positive effect on the intake and the rate of food passage through the digestive tract and improved the absorption of soluble carbohydrates.

In this study, a significant reduction in total cholesterol (48.37 mg/dL) until the age of 77 d was observed in Nanosoy-diet-fed rabbits as compared to that in Nanopupunha-diet-fed animals (98.50 mg/dL). The presence of gel-forming NSPs (d-xylose and d-arabinose) in the SI reduces starch digestion and glucose absorption [60]. Soluble NSPs present in the cell walls of soybean hulls, with the ability to retain water and form gels, have low digestibility and undergo rapid fermentation in the LI [43]. The increase in viscosity in the intestinal lumen reduces the digestion and absorption of lipids and the solubilization of bile salts, which alters the metabolic pathways of hepatic cholesterol and lipoproteins and results in the reduction of plasma cholesterol [61,62]. This mechanism could explain the lower cholesterol levels in animals that received Nanosoy in the diet.

According to DeLoid et al. [7], nanofibers can also bind more easily to sugars and fats in the SI, making them less available for digestion and reducing blood cholesterol levels. In this way, the inclusion of Nanopupunha can improve the use of nutrients and lipids in the diet, prevent intestinal disorders, and improve the nutritional efficiency and health of growing rabbits.

The IgM and IgG values of the animals fed different experimental diets were statistically equal. However, these values were higher than those described by Zhu et al. [63], with IgG values detected in the blood serum of rabbits at the age of 82 d ranging from 1.46–1.55 mg/mL. Owing to their greater absorption/interaction capacity, nanofibers can penetrate the mucus layer of the SI [64]. In this way, they can activate inflammasomes that signal the activation of IL-1b [65], thereby stimulating the non-specific immune response [66] and the ability to adapt to the immune system [67] in response to different types of dietary fiber and metabolic signals in the SI and LI.

### 4.6. Cecal Bacterial Count

Nanofibers improved the cecal microbiota count and helped in the multiplication of beneficial bacteria (lactic-acid). The lactic-acid bacterial count was higher (*p* < 0.05) in rabbits fed Nanosoy- (6.83 log CFU/g) and Nanopupunha-supplemented (5.71 log CFU/g) diets compared to those fed control diet (2.59 log CFU/g). The animals fed with the Nanopupunha and control diets had lower counts of Enterobacteriaceae than those fed Nanosoy. Chen et al. [53] reported that Lactobacillus and Bifidobacterium counts were higher in piglets fed non-gel-forming fibers, and Enterobacteriaceae counts were higher in animals fed a diet with soybean hulls. Thus, the chemical nature of fibers, particularly their fermentability, modulates microbial activity [68], maintains the balance and diversity of the intestinal microbiota of rabbits [28], and provides better intestinal health for the animals.

Digestible insoluble cell wall NSPs (cellulose, hemicellulose, and lignin) act as substrates for microbial fermentation in the gastrointestinal tract and play an important nutritional role by synthesizing SCFA [38]. SCFAs participate in microbiota modulation and reduce immune stress and low immunity [69,70], as observed in the present study with the inclusion of Nanopupunha in the diet of growing rabbits. NSP digestion stimulates the increase of TGF-α, which maintains the integrity of the intestinal mucosa [52] and promotes a positive effect on intestinal health, improves cecal microbiota in rabbits, and makes better use of the energy from the diet.

## 5. Conclusions

The inclusion of 7% nanofibers from the peach palm sheath into the diet improved the intestinal health of growing rabbits. In addition, it did not alter zootechnical performance and did not promote physiological or clinical changes or show deleterious effects until the animals were 77 days old. Nanofibers are a promising ingredient for animal nutrition and allow for a better method to introduce nutrients into the diet and promote more sustainable methods of animal production.

## Figures and Tables

**Figure 1 animals-13-00192-f001:**
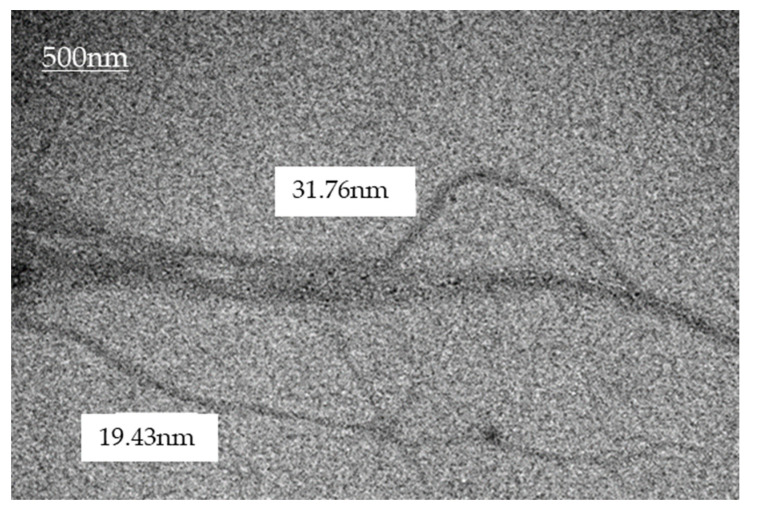
Characterization of nanofibers through Transmission Electron Microscopy (TEM).

**Figure 2 animals-13-00192-f002:**
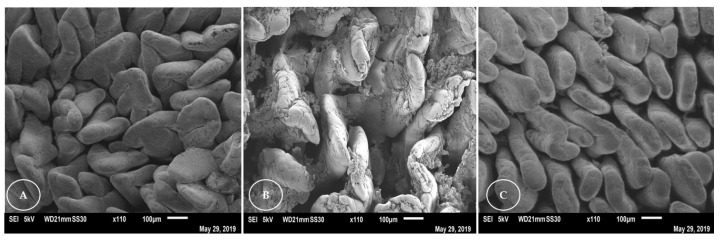
Ultrastructural analysis of the duodenum of rabbits that were fed three experimental diets: (**A**) Control, (**B**) Nanosoy, and (**C**) Nanopupunha.

**Figure 3 animals-13-00192-f003:**
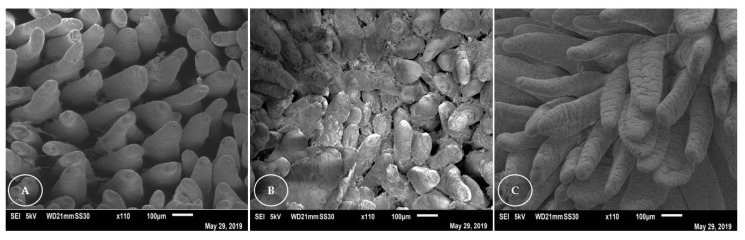
Ultrastructural analysis of the jejunum of rabbits that were fed three experimental diets: (**A**) Control, (**B**) Nanosoy, and (**C**) Nanopupunha.

**Figure 4 animals-13-00192-f004:**
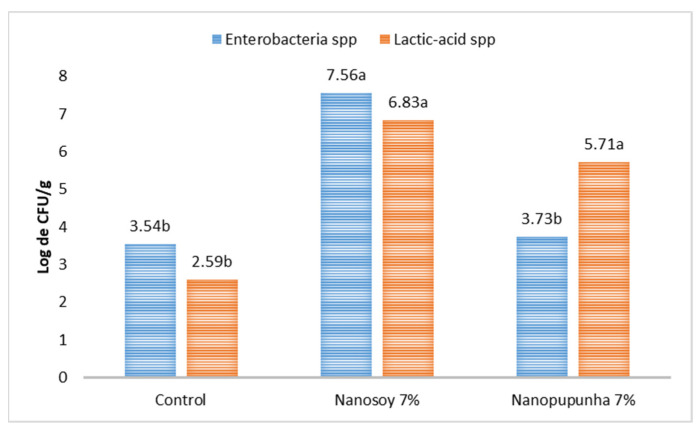
Count of Enterobacteriaceae spp. and lactic-acid bacteria (log CFU/g) in the cecum of growing rabbits fed different diets viz., Control, Nanosoy, and Nanopupunha. Lowercase letters differ statistically based on Tukey’s test (*p* < 0.05).

**Table 1 animals-13-00192-t001:** Compositions and the calculated and analyzed nutritional values of the three experimental diets fed to growing rabbits.

Ingredients (%)	Control	Nanosoy	Nanopupunha
Corn, 8% CP	16.83	24.48	24.48
Wheat bran, 16% CP	29.50	31.50	31.50
Soybean meal, 45% CP	23.00	23.50	23.50
Alfalfa dehydrated	8.50	2.00	2.00
Soybean oil	2.50	-	-
Nanofiber gel	-	10.50	10.50
Molasses aroma	0.02	0.02	0.02
Rice husk	15.65	4.00	4.00
^1^ Premix Nuvilab rabbits	4.00	4.00	4.00
Total	100.00	100.00	100.00
Calculated nutritional values			
Dry matter, %	90.12	90.31	90.31
Crude protein, %	18.31	18.05	18.05
Ethereal extract, %	5.07	2.70	2.70
Crude energy, Kcal/Kg	3.180	3.180	3.180
Crude fiber, %	14.00	14.00	14.00
Nanofibers, %	-	7.00	7.00
Lysine, %	0.96	0.95	0.95
Methionine, %	0.30	0.31	0.31
Calcium, %	1.08	1.00	1.00
Total phosphorus, %	0.63	0.65	0.65
Nutritional values analyzed			
ADF, %	15.50	12.30	11.70
NDF, %	28.50	24.30	25.20

Crude protein (CP), crude fiber (CF), acid detergent fiber (ADF), and neutral detergent fiber (NDF). ^1^ Mineral and Vitamin supplement composition per kg of feed: vitamin A (min), 8000 uI/kg; vitamin D3 (min), 1200 uI/kg; vitamin E (min), 20 uI/kg; vitamin K3 (min), 1 mg/kg; vitamin B1 (min), 2 mg/kg; vitamin B2 (min), 6 mg/kg; vitamin B6 (min), 2 mg/kg; vitamin B12 (min), 10 mcg/kg; nacin (min), 30 mg/kg; pantoten calcium (min), 17 mg/kg; folic acid (min), 1 mg/kg; biotin (min), 0.03 mg/kg; choline (min), 1400 mg/kg; sodium minerals (min), 2700 mg/kg; iron (min), 40 mg/kg; manganese (min), 40 mg/kg; zinc (min), 60 mg/kg; copper (min), 6 mg/kg; iodine (min), 0.3 mg/kg; selenium (min), 0.1 mg/kg; cobalt (min), 1 mg/kg; fluorine (max), 60 mg/kg; amino acid methionine (min), 2700 mg/kg; lysine (min), 8000 mg/kg; BHT additives, 100 mg/kg.

**Table 2 animals-13-00192-t002:** Zootechnical performances of growing rabbits fed different sources of nanofibers and those fed a control diet.

Variables ^1^	Treatments			SEM ^2^	*p*-Value
	Control	Nanosoy 7%	Nanopupunha 7%		
IBW, kg	0.939	0.912	0.946	0.035	0.923
FBW, kg	2.771 ^a^	2.292 ^b^	2.770 ^a^	0.074	0.001
DFI, kg	0.157 ^a^	0.111 ^b^	0.149 ^a^	0.004	0.001
DBWG, kg	0.044 ^a^	0.033 ^b^	0.043 ^a^	0.001	0.001
FCR	3.606	3.427	3.443	0.079	0.632

^1^ Averages of litter body weight (IBW) and final body weight (FBW), daily feed intake (DFI), daily body weight gain (DBWG), and feed conversion ratio (FCR). ^2^ SEM: Standard error of the mean. ^a,b^: Distinct letters on the same line indicate significant differences based on Tukey’s test (*p* < 0.05).

**Table 3 animals-13-00192-t003:** Means of the relative weights of digestive tract organs and appendages of growing rabbits fed different sources of nanofibers and those fed a control diet.

Variables ^1^	Treatments			SEM ^2^	*p*-Value
	Control	Nanosoy	Nanopupunha		
FBW, kg (77 day)	2.771 ^a^	2.292 ^b^	2.770 ^a^	0.074	0.001
Stomach, %	4.454 ^a^	4.174 ^ab^	4.079 ^b^	0.097	0.009
SI, %	2.338 ^c^	2.587 ^b^	2.770 ^a^	0.066	0.001
LI, %	10.106 ^a^	9.765 ^a^	9.122 ^b^	0.224	0.037
Liver, %	5.105 ^a^	4.420 ^b^	5.110 ^a^	0.125	0.001
Kidneys, %	0.715 ^b^	0.788 ^a^	0.786 ^a^	0.017	0.025
Spleen, %	0.039 ^a^	0.028 ^b^	0.039 ^a^	0.001	0.001

^1^ Average weights of the stomach, small intestine (SI), large intestine (LI), liver, kidneys, and spleen, depending on the diets. ^2^ SEM: Standard error of the mean. ^a,b^: Different letters on the same line indicate statistically significant differences based on Tukey’s test (*p* < 0.05).

**Table 4 animals-13-00192-t004:** Mean pH values of the digestive tract contents obtained from growing rabbits fed different sources of nanofibers and those fed a control diet.

Variables ^1^	Treatments			SEM ^1^	*p*-Value
	Control	Nanosoy	Nanopupunha		
Stomach pH	2.868 ^a^	2.141 ^b^	2.750 ^a^	0.103	0.012
Jejunum pH	7.828 ^a^	6.885 ^b^	7.846 ^a^	0.105	0.001
Cecum pH	6.940 ^ab^	6.479 ^b^	7.269 ^a^	0.128	0.048

^1^ SEM: Standard error of the mean. ^a,b^: Different letters on the same line indicate statistically significant differences by Tukey’s test (*p* < 0.05).

**Table 5 animals-13-00192-t005:** Structural analysis and villi density of the intestinal epithelium of growing rabbits fed different sources of nanofibers and those fed a control diet.

Variables ^1^	Treatments			SEM ^2^	*p*-Value
	Control	Nanosoy	Nanopupunha		
Duodenum					
VH (µm)	709.620 ^b^	939.608 ^a^	964.772 ^a^	40.149	0.009
CD (µm)	79.986 ^b^	172.308 ^a^	131.291 ^ab^	12.851	0.021
VH:CD	9.688	5.874	8.445	0.682	0.110
VD *	24.365 ^b^	27.417 ^ab^	31.541 ^a^	1.247	0.027
VW (µm)	114.616 ^b^	149.888 ^a^	151.255 ^a^	7.26	0.043
TMT (µm)	847.941 ^b^	1245.555 ^a^	1203.059 ^a^	55.128	0.002
WT (µm)	47.122 ^b^	81.605 ^ab^	99.069 ^a^	7.102	0.007
Jejunum					
VH (µm)	710.481	726.017	787.724	37.154	0.636
CD (µm)	87.862 ^b^	139.751 ^a^	89.534 ^ab^	8.932	0.028
VH:CD	8.522 ^a^	5.922 ^b^	9.013 ^a^	0.513	0.011
VD *	24.361	31.042	37.968	2.813	0.096
VW (µm)	120.854	142.327	115.213	6.900	0.195
TMT (µm)	816.190	924.502	896.674	40.567	0.542
WT (µm)	79.857	116.983	82.812	7.544	0.091

^1^ Villus height (VH), crypt depth (CD), VH/CD ratio, villus density (VD), villus width (VW), total mucosal thickness (TMT), and wall thickness (WT) of the duodenum and jejunum of growing rabbits; ^2^ SEM—Standard error of the mean; ^a,b^: Different letters, on the same line, differ statistically based on Tukey’s test (*p* < 0.05). * VD: Density of villi = number of villi/922 μm^2^.

**Table 6 animals-13-00192-t006:** Plasma glucose levels in rabbits at the beginning of the experiment (age = 35 days) and the glucose, cholesterol, and triglyceride (mg/dL) levels at the end of the experiment (age = 77 days).

Variables (mg/dL)	Treatments			SEM ^1^	*p*-Value
	Control	Nanosoy	Nanopupunha		
Day 35					
Blood glucose	129.500 ^B^	158.625 ^A^	123.375 ^A^	11.905	0.445
Day 77					
Blood glucose	169.500 ^A^	191.625 ^A^	153.125 ^A^	8.866	0.211
Cholesterol	70.125 ^ab^	52.429 ^b^	98.500 ^a^	7.256	0.007
Triglycerides	104.625	76.250	122.875	10.665	0.224

^1^ SEM—Standard error of the mean. ^a,b^: Different lowercase letters on the same line differ statistically based on the Tukey’s test (*p* < 0.05). ^A,B^: Different capital letters in the same column indicate a significant difference between blood glucose levels on days 35 and 77 using Tukey’s test (*p* < 0.05).

**Table 7 animals-13-00192-t007:** Immunoglobulin (IgM and IgG) concentrations in the blood serum of growing rabbits fed different sources of nanofibers and those fed a control diet.

Variables	Treatments			SEM ^1^	*p*-Value
	Control	Nanosoy	Nanopupunha		
IgM (mg/mL)	0.626	0.558	0.619	0.021	0.391
IgG (mg/mL)	5.741	4.858	6.153	0.335	0.285

^1^ MSE—Mean Standard Error.

## Data Availability

Not applicable.

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
