# Peer review of "Inclusion of Soybean Hulls (Glycine max) and Pupunha Peach Palm (Bactris gasipaes) Nanofibers in the Diet of Growing Rabbits: Effects on Zootechnical Performance and Intestinal Health"

_animals, 2023, doi:10.3390/ani13020192_

Round 1
Reviewer 1 Report
Minor corrections pointed out in the attached file should be made before acceptance
References must be more recent

Author Response
Review 1
Dear Reviewer, thanks for contributions
- Minor corrections pointed out in the attached file should be made before acceptance
Answer: Some changes were made to section L17: Allocaated was changed to allocated
Answer: Some changes were made to section L165-L166: “In the jejunum, an incision was made in the median portion of the cecum and the pH was measured” was changed to “In the jejunum, an incision was made in the median portion, and in the cecum, an incision was made in the median portion for the pH measured.”
- References must be more recent
Answer: Some changes were made to sections References: L563 – L730

Reviewer 2 Report
In my opinion, manuscript entitled ,,Inclusion of Soybean Hulls (Glycine max) and Pupunha Peach Palm (Bactris gasipaes) Nanofibers in the Diet of Growing Rabbits: Effects on Zootechnical Performance and Intestinal Health” is valuable, although I am not entirely satisfied with the design. Well, it seems to me that the authors could have used a bit more animals, at least n=10 and one sex (in this type of research usually males are used). I would like to point out, however, that this is basically my only substantive objection to this work and only it could possibly decide whether or not to accept a manuscript for publication in Animals. Below are some of my minor observations:
- Table 2: Instead of IW and FW authors should use terms: IBW and FBW or BW 35 d and BW 77 d. The same applies to DBWG.
- Table 6: Blood glucose 35 d and 77 d. If overal p>0.05 (the right column) than you should not consider between-groups differences as significant.
- Conclusions: The last sentence (,,Nanotechnology is a promising area… of animal production”) is more of a didactic conclusion than it results from the results obtained in the work, which is why I propose to remove it.
Author Response
Review 2
Dear Reviewer, thanks for contributions
In my opinion, manuscript entitled, “Inclusion of Soybean Hulls (Glycine max) and Pupunha Peach Palm (Bactris gasipaes) Nanofibers in the Diet of Growing Rabbits: Effects on Zootechnical Performance and Intestinal Health” is valuable, although I am not entirely satisfied with the design. Well, it seems to me that the authors could have used a bit more animals, at least n=10 and one sex (in this type of research usually males are used). I would like to point out, however, that this is basically my only substantive objection to this work and only it could possibly decide whether or not to accept a manuscript for publication in Animals. Below are some of my minor observations:
Answer: In order to carry out this study, a number of animals (n8) were used, considering the production capacity of Nanofibers, as well as the preparation of the feed for the experimental period. We observed that the Nanofibers gel needed to be stored at a temperature of 10°C and incorporated into the diet for a maximum period of 12 hours, due to the high fermentation process of the Nanofibers. For this, the statistical power was calculated: The sample size was calculated to the Test-T of Student considering a difference between the groups (Alpha= 5%), statistical power of 80% and maximum of error margin for this study of 10%.
- Table 2: Instead of IW and FW authors should use terms: IBW and FBW or BW 35 d and BW 77 d. The same applies to DBWG.
Answer: Table 2 - Some changes have been made to table and legends: L255-257 - IW, kg change to IBW, kg; FW, kg change to FBW, kg; DWG change to DBWG, kg.
- Table 6: Blood glucose 35 d and 77 d. If overal p>0.05 (the right column) than you should not consider between-groups differences as significant.
Answer: Some changes were made to section L317-319 - A, B Different capital letters in the same column indicate a significant difference between blood glucose levels on days 35 and 77 using Tukey's test (p < 0.05).
Conclusions: The last sentence (“Nanotechnology is a promising area… of animal production”) is more of a didactic conclusion than it results from the results obtained in the work, which is why I propose to remove it.
Answer: Some changes were made to 5. Conclusions: L527-532 - (… Nanofibers is a promising ingredient for animal nutrition and allows a better method to introduce nutrients into the diet and promote more sustainable methods of animal production).

Reviewer 3 Report
I have reviewed the manuscript entitled “Inclusion of Soybean Hulls (Glycine max) and Pupunha Peach Palm (Bactris gasipaes) Nanofibers in the Diet of Growing Rabbits: Effects on Zootechnical Performance and Intestinal Health” submitted for a publication in Animals as an original article. However, Authors didn't draft the paper well and I cannot recommend the publication of this form.
1. In general, materials and methods section in poorly written, with a large amount of missing information or inaccuracies. For wxample: how nanofiber structure was measured? distance between disc equal 0.1 mm (100 um) is not sufficient to produce nanofiber (L86) .How gel composition was determined? (L86 and L 89)
2. Diet composition is not clearly presented. How hanosoy diet containing 10.5% of nanofiber gel can contain 7% of nanofibers as the nanofiber gel contains only 7% of itself ? The actual concentration of nanofibers in diets would be 0.73% in that case.
3. Organ weight/relative weight is not a measure of organ morphometry. Morphometry refers to the quantitative analysis of form.
4. Please revise the abbreviation used – different abbreviation of intestinal morphology parameters are introduced in methods section (2.6), different are used in Table 5.
5. Discussion should be corrected. In general, avoid repeating the exact results in discussion. Some scctions should be rephased – for example sections from L346-351 and L363-367 – could be merged ,as these paragraphs repeats the same information.
6. 2. References must be verified. For example, what is the connection of building materials with animal sciences (L37) ? Ref. [3] is out of topic, Ref [6] in L42 is incorrect.
7. English should improve by a native person.
8. Simple summary is missing
Minor (selected) comments:
L12 correct
L37 “Nanofibers are stable atomic structures” – please rephase
L62 “organ morphometry” – pleas be more precise
Table 1 – correct comma as decimal separator to dot
All bacteria species should be written italic.
Figure 3 – Correct to English in X and Y axes
Author Response
Review 3
Dear Reviewer, thanks for contributions
- In general, materials and methods section in poorly written, with a large amount of missing information or inaccuracies. For example: how nanofiber structure was measured? distance between disc equal 0.1 mm (100 um) is not sufficient to produce nanofiber (L86).
Answer: Characterization of the nanofibers: L97-108
“The nanofiber were characterized in terms of their dimensions and chemical characteristics. For the analysis in Transmission Electron Microscope (TEM), the sample was submitted to a procedure of individualization of the cellulose fibers. For this process, the sample was diluted in 4 parts of PA ethyl alcohol in eppendordf. This diluted sample was subjected to sonication for 60 min. After this step, the sample was dripped onto a screen covered with Parlodio. Once dried at room temperature, the samples were analyzed using a Transmission Electron Microscope (TEM), brand JEOL, model JEM1200EX-II, located at the Center for Electronic Microscopy (CME) of UFPR. The resulting images were processed with the software Paint.net TM version 3.5.10, which allowed an estimation of the fibril dimensioning (Figure 1).
Figure 1. Characterization of nanofibers through Transmission Electron Microscopy (TEM)
- How was gel composition determined? (L86 and L 89).
Answer gel composition: L108-114
“The bromatological analyzes of the vegetable raw materials (soybean hulls, pupunha peach palm sheath) and the respective nanofiber gels were performed at the Nutrition Laboratory at PUCPR and expressed in % based on dry matter (m/m). For weighing the samples, an analytical precision balance was used (Mars, AY220). The total fiber content was determined using the enzymatic-gravimetric method modified according to the Chemical and Physical Methods methodology for food analysis of the Analytical Norms of the Instituto Adolfo Luz (Brasil, 2005).
- Diet composition is not clearly presented. How nanosoy diet containing 10.5% of nanofiber gel can contain 7% of nanofibers as the nanofiber gel contains only 7% of itself? The actual concentration of nanofibers in diets would be 0.73% in that case.
Answer: L116-120 - The inclusion of Nanosoja and Nanopupunha gel with 7% of Nanofibers in the natural matter was calculated to obtain an increment of 7% of nanofibers in the dry matter of the feed, considering the formula: Suspended mass of nanofibers (MSN) = Final mass of feed (MF) × percentage of nanofibers in the feed (NF) ÷ percentage of nanofibers in the gel (NG).
- Organ weight/relative weight is not a measure of organ morphometry. Morphometry refers to the quantitative analysis of form.
Answer: Some changes were made to topic 3.2 Morphometry of organs was change to Relative organ weight
- Please revise the abbreviation used – different abbreviation of intestinal morphology parameters are introduced in methods section (2.6), different are used in Table 5.
Section (2.6) villus height (AV), crypt depth (PC), villus width (LV; measured at one-third of the villus base), full thickness of the mucosa (ETM) and wall thickness (PE), and calculation of the villus height/crypt depth (AV/PC).
Answer: Some changes were made to section 2.7 and table 5: L175-175 - Villus height (VH), crypt depth (CD), VH/CD ratio, villus density (VD), villus width (VW), total mucosal thickness (TMT) and wall thickness (WT).
- Discussion should be corrected. In general, avoid repeating the exact results in discussion. Some sections should be rephased – for example sections from L346-351 and L363-367 – could be merged, as these paragraphs repeats the same information.
- Some changes were made to sections L386-391: In our study, the inclusion of 7% Nanosoy in the diet resulted in greater weight of the LI compared to that in the Nanopupunha-fed animals. Arruda et al. [37] reported higher cecal weight and content in animals fed soybean hulls. The soybean hull fibers — pectin and oligosaccharides — reduces digestion of the diet in the SI, thereby contributing to an increase in the volume of digesta in the cecum [30,33,37,]. This results in the greater development of the cecum in rabbits because of an increase in excreta production and a higher rate of microbial fermentation.
- Some changes were made to sections L402-408: The lower hepatic weight in the Nanosoy-fed animals was due to the chemical structure of the soybean hull fibers, which promoted a lower absorption capacity of the dietary and negatively influenced the liver weight in growing rabbits. Soluble fibers in the soybean hull stimulate an increase in digesta viscosity, which reduces the action of lipase [42], leading to changes in lipid absorption and metabolism [31,33]. These factors affect triglyceride content in the liver [43] as well as cholesterol levels (Table 6) in the blood serum, which may favor lipogenesis in adipocytes.
- 2. References must be verified. For example, what is the connection of building materials with animal sciences (L37)? Ref. [3] is out of topic, Ref [6] in L42 is incorrect.
Answer: Some changes were made to references [3], Ref [6] in the sections (L37) and (L42)
- English should improve by a native person.
Answer: The manuscript was forwarded to a publishing company (Editage) for a full review in English by a native speaker. Attached is the revision certificate issued by Editage.
- Simple summary is missing
Answer:
Simple Summary: This estudy evaluated the inclusion of nanofibers from soybean hulls and pupunha peach palm heart sheaths in the diet of growing rabbits. The results reveal, although the inclusion of 7% Nanopupunha in the diet of rabbits did not alter the performance, it improved intestinal health and increased the lactic-acid bacterial count in the cecum of growing rabbits. Therefore, the inclusion of nanometric particles of plant fibers (<100 nm) in animal feed can be used as potential to fiber alternative and provide excellent physical characteristics for improved nutrient absorption from the diet.
Minor (selected) comments:
L12 correct
L37 “Nanofibers are stable atomic structures” – please rephase
Answer: Some changes were made to sections L41-42 “Nanofibers are stable atomic structures with biodegradable characteristics and low-production costs [3,4].
L62 “organ morphometry” – pleas be more precise
Answer: Some changes were made to sections L69: Organ morphometry change to Relative organ weight
Table 1 – correct comma as decimal separator to dot
Answer: Some changes were made to decimal separator in Table 1
All bacteria species should be written italic.
Answer: In the present study, only the population of enterobacteria belonging to the Gram-negative and Gram-positive bacilli family was enumerated. No specific bacterial species were identified.
Figure 3 – Correct to English in X and Y axes
Answer: Some changes were made to English in X and Y axes

Round 2
Reviewer 3 Report
The Authors have answered all my main issues which were raised during first review.